



**HESS Opinions: Beyond the Long-term Water Balance: Evolving Budyko's**
**Legacy for the Anthropocene towards a Global Synthesis of Land-surface**
**Fluxes under Natural and Human-altered Watersheds**
**A. Sankarasubramanian[1], Dingbao Wang[2], Stacey Archfield[3], Meredith Reitz[3],**
**Richard M. Vogel[4], Amirhossein Mazrooei[1] and Sudarshana Mukhopadhyaya[1]**
[1]Department of Civil, Construction and Environmental Engineering, North Carolina State
University, Raleigh, NC 27695.
[2]Department of Civil, Environmental, and Construction Engineering, University of Central
Florida, Orlando, FL 32816.
[3]Water Mission Area, U.S. Geological Survey, Reston, VA 20192.
[4]Department of Civil and Environmental Engineering, Tufts University, Medford, MA 02155.
**Abstract**
Global hydroclimatic conditions have been significantly altered over the past century by
anthropogenic influences that arise from the warming global climate and also from local/regional
anthropogenic disturbances. Traditionally, studies have used coupling of multiple models to
understand how land-surface fluxes vary due to changes in global climatic patterns and local
land-use changes. We argue that Budyko's framework that relies on the supply and demand
concept could be effectively adapted and extended to quantify the role of drivers – both changing
climate and local human disturbances – in altering the land-surface response across the globe.
We review the Budyko framework along with potential extensions with an intent to further the




applicability of the framework to emerging hydrologic questions.  Challenges in extending the
Budyko framework over various spatio-temporal scales and evaluating the water balance at these
various scales with global data sets are also discussed.

**28    The historical evolution of the Budyko framework in hydroclimatology**

The traditional Budyko formulation provides the long-term water balance as a single-

stage partitioning of precipitation into runoff and evapotranspiration; and it has been verified
over thousands of natural watersheds around the globe (Zhang et al., 2004; Yang et al., 2007;
Sivapalan et al., 2011; Williams et al., 2012; Padrón et al., 2017). Besides the aridity index,
which is defined as the ratio of the mean annual potential evapotranspiration to the mean annual
precipitation, Milly et al. (1994) and Sankarasubramanian and Vogel (2002) proposed additional
controls on the long-term water balance including seasonality and soil moisture holding capacity
that enhanced the Budyko framework for explaining the spatial variability in mean annual runoff
at the continental scale. Studies have also extended the Budyko framework for capturing the
interannual variability in runoff (Koster and Suarez, 1999; Sankarasubramanian and Vogel,
2002, 2003).  More recently, the Budyko framework has been extended for explaining the
seasonal hydroclimatology of basins (Petersen et al., 2012; Chen et al. 2013; Petersen et al.
2018).  Similarly, the Budyko framework has been extended for quantifying the non-dimensional
sensitivity (also termed elasticity) of land-surface response to changes in climatic controls under
different hydroclimatic regimes (Dooge, 1992; Dooge et al., 1999; Sankarasubramanian et al.,

2001).

Perhaps the most unique aspect of the Budyko framework lies in its Darwinian approach

which enables us to view the entire hydroclimatic system without focusing on each physical





process in isolation (Harman and Troch, 2014; Wang and Tang, 2014). Darwinian approach
seeks to document patterns of variation in populations of hydrologic systems and develop
theories that explain these observed patterns in terms of the mechanisms and conditions that
determine their historical development (Harman and Troch, 2014).  Even though most studies
which employed Budyko's framework have focused on natural basins, the original monograph
(Budyko, 1974), Climate and Life, considered the role of human influence on climate including
impacts of reservoir storage and irrigation on evapotranspiration. As hydroclimatic regimes
evolve in the Anthropocene, it is critical to understand how land-surface fluxes change due to
changes in local watershed conditions and due to global climate change.  Given the Budyko
framework's emphasis on a Darwinian approach and its ability to capture the fundamental
dimensions of land-surface fluxes, a global synthesis on the variability in these fluxes across
natural and human-altered watersheds should provide insights on the sensitivity of the critical
hydroclimatic processes to local and global changes in the Anthropocene.

**Budyko Framework for the Anthropocene**

We are at a critical time in which the hydroclimate, particularly land-surface fluxes, has

been significantly altered over the past century by anthropogenic disturbances (Entekhabi et al.,
1999; Vogel et al., 2015). For instance, both annual precipitation and streamflow have increased
during the period of 1948–1997 across the eastern United States, and those trends appear to arise
primarily from increases in autumn precipitation (Small et al., 2006; Rice et al., 2015). Similarly,
the frequency of floods is increasing in many regions, while magnitudes of flooding appear only
to be systematically increasing in certain spatially cohesive regions (Hirsch and Archfield, 2015;
Malikpour and Villarini, 2015; Archfield et al., 2016) particularly in urban areas (Vogel et al.,





2011; Barros et al., 2014 and Prosdocimi et al., 2015).  Irrigation in the U.S. high plains leads to
increases in summer rainfall and streamflow in the Midwest due to land-surface and atmosphere
feedback (Kustu et al., 2011). Based on hydroclimatic observations from 100 large hydrological
basins globally, Jaramillo and Destouni (2015) found consistent and dominant effects of
increasing relative evapotranspiration from flow regulation and irrigation and decreasing
temporal runoff variability from flow regulation. Development of irrigation networks and man-
made reservoirs also increased surface water and groundwater withdrawals and land-use changes
(Maupin et al., 2014; Sankarasubramanian et al., 2017; Das et al., 2018). Similarly, construction
of large dams has significantly altered the downstream flow variations impacting downstream
ecology (Gao et al., 2008; Wang et al., 2017). Changes in land-use and land-cover also impact
the local energy balance creating urban heat islands (Memon et al., 2008), affecting recharge and
baseflow (Price, 2011), which in turn impacts a very broad range of streamflows (Allaire et al.,
2015) with particularly significant increases in high flows (Vogel et al., 2011; Barros et al. 2014;
Prosdocimi et al. 2015). Thus, anthropogenic influences arising from global climate change and
local to regional disturbances can significantly impact the land-surface response from the
watershed. Anthropogenic influences including changes in climate, land use, and water use
exhibit complex interactions which must be considered jointly, to understand their impact on
hydrologic flow alteration (Allaire et al. 2015).  Performing a synthesis on how the spatio-
temporal variability of land-surface fluxes – runoff, evapotranspiration, net radiation, and
hydrologic flow alteration – differ globally in natural and human-altered watersheds is a critical
need to enable a complete understanding of global hydroclimate during the Anthropocene.  The
Budyko framework provides an ideal approach for such inquiry, because it has been used to
decompose changes in long-term land-surface fluxes due to both natural variability and human



influence (e.g., Roderick and Farquhar, 2011; Wang and Hejazi, 2011; Yang et al., 2014; Jiang et
al., 2015).
**Budyko Framework Adaptation in Watershed Modeling**

Figure 1 provides the general setup of the Budyko framework to explain the spatio-

temporal variability of land-surface fluxes in natural watersheds and human-altered landscapes.
The framework relies on conservation of mass and energy to model and predict the "actual"
hydroclimatic variable of interest based on the available "demand" and "supply" of mass and
energy (Figure 1). The rationale for using the Budyko framework for understanding the spatial
variability in land-surface fluxes over natural/human-altered watersheds lies in its ability to
capture the hydroclimatic dimensions of supply and demand, thereby providing a low-
dimensional parsimonious approach (Figure 1) to this multidimensional problem.  Here, we
evaluate and extend the Budyko framework for understanding the spatio-temporal variability of
different land-surface fluxes.
*Long-term Water Balance*

The most commonly used framework for modeling long-term water balance is to estimate

the mean annual evapotranspiration ("actual") based on the ratio of mean annual potential
evapotranspiration ("demand") to the mean annual precipitation ("supply"). Thus, the upper limit
for mean annual evapotranspiration is potential evapotranspiration (precipitation) in a humid
(arid) region. The family of Budyko curves estimates the evapotranspiration ratio ("actual"/
"supply") based on the aridity index ("demand"/"supply").  For additional details, see
Sankarasubramanian and Vogel (2001). Most studies have focused on evaluating the long-term
water balance at regional and continental scale (see Wang et al., 2016 for a detailed review).
Studies have also focused on the impact of land cover and climate on long-term water yield using



global data (Zhou et al., 2015). Here, we evaluate the Budyko framework to the global scale
using the data from the Global Land-Surface Data Assimilation System, version 2 (GLDAS2)
(Rodell et al., 2004).  Data points of mean annual evapotranspiration and aridity index are
obtained from the GLDAS2 dataset with a spatial resolution of 0.25 $^{\circ}$ for the period 1948-2010.
Figure 2 shows the performance of the Budyko curve in estimating the mean annual
evapotranspiration based on the aridity index data between 60º S to 60º N.  Even though the
Budyko curve provides a first-order approximation of the spatial variability in the
evapotranspiration ratio (Figure 2), the scatter around the curve is quite considerable. Studies
have shown that seasonality in moisture and energy and their co-availability (i.e., phase
difference between moisture and energy availability within the year) and soil moisture holding
capacity partially control the scatter around the Budyko curves (Milly et al., 1994,
Sankarasubramanian and Vogel, 2003). Another question of interest is to understand the lower
bound on the evapotranspiration ratio, which is typically limited by the moisture availability in a
region (Wang and Tang, 2014). Numerous studies on long-term balance have employed fitting
the observed long-term water balance by parameterizing the Budyko curves (see Wang et al.,
2016 review paper). However, limited/no effort has been undertaken on how this data cloud of
long-term water balance cloud is expected to change under potential climate change and how this
interplay between moisture and energy is expected to affect the long-term water balance under
different type of watersheds (Creed et al., 2014). Similarly, recent studies have extended
Budyko's steady-state supply-to-demand framework for modeling land-surface fluxes over fine
(daily and monthly) time scales (Zhang et al., 2008). Validating these emerging frameworks with
global hydrologic data will provide an understanding of the critical process controls in estimating
land-surface fluxes.  This validation effort will also help in understanding the advantages and



limitations of such parsimonious modeling approach towards estimating evapotranspiration and
streamflow at various spatio-temporal scales.
***Extension of Budyko's "supply and demand" concept for infiltration***

The upper bounds on the Budyko framework arise from the conservation of mass and

energy. Hence, in principle, it could be applied to other hydrological processes. Zhang et al.
(2008) applied the Budyko's monthly supply and demand attributes to estimate the catchment
retention and the overland runoff from the soil moisture zone.  Wang (2018) developed the
infiltration equation for saturation excess in the Budyko's supply and demand framework, i.e.,
modelling the ratio of infiltration to rainfall depth as a function of the ratio between infiltration
capacity and rainfall depth (Figure 3).  The cumulative infiltration depth during a rainfall event is
defined as the "actual" variable of interest, and the cumulative rainfall depth during an event is
defined as the "supply".  The effective soil water storage capacity for the event is defined as the
"demand", which is dependent on the initial soil moisture condition.  In Figure 3, the initial soil
moisture condition is represented by the degree of saturation, $\psi$, which is defined as the ratio of
initial soil water storage and storage capacity (Wang, 2018).  For a dry soil with low $\psi$,
infiltration is expected to be higher with lower surface runoff potential. The upper bounds of
these curves (Figure 3) are similar to the Budyko's asymptotes corresponding to infiltration
capacity-limited and rainfall depth-limited conditions. In this illustration, the Budyko framework
is extended to estimate the temporal variability of infiltration into the soil based on soil water
storage capacity and antecedent conditions ($\psi$). Thus, the parsimonious framework stems from
the Budyko's supply and demand concept to develop the asymptotes and then use those
asymptotes to identify and explain various critical process controls (e.g., infiltration in Figure 3).

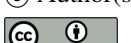



161   Although the above extensions of the Budyko framework demonstrate the potential for

162 developing a low-dimensional parsimonious modeling strategy, data-based validation efforts

163 have focused primarily on the long-term hydroclimatic attributes (i.e., mean, variance and

164 elasticity) of observed land-surface fluxes in natural basins (Figure 2) (Sankarasubramanian and

165 Vogel, 2001; Abatzoglou and Ficklin, 2017). Representing a hydroclimatic variable of interest

166 (i.e., "actual") as a ratio to the "supply" and explaining its spatio-temporal variability based on

167 the demand/supply ratio and other variables (e.g., soil moisture holding capacity for long-term

168 water balance) provides a simplistic, non-dimensional form for understanding the process

169 controls. For instance, in the long-term water balance context, defining the demand/supply

170 relationship explains the predominant controls on the spatio-temporal variability of mean annual

171 runoff and mean annual evapotranspiration based on the basin aridity, seasonality of demand and

172 supply (i.e., in-phase or out-of-phase between moisture and energy) attributes and soil moisture

173 holding capacity (Milly, 1991). Synthesizing relevant process controls and representing them

174 within the Budyko low-dimensional framework will also help us in the catchment classification

175 and in understanding how different hydroclimatic processes of interest vary across wider regimes

176 and landscapes.

178 **Extending Budyko Framework for Human-altered Watersheds and Landscapes**

179  Figures 4-6 extend the Budyko framework to explain the spatio-temporal variability in

180 land-surface fluxes in human-altered watersheds and landscapes. A synthesis involving extension

181 and evaluation of the Budyko framework for estimating land-surface fluxes in human-altered

182 watersheds will help us understand the role of key drivers and anthropogenic disturbances (e.g.,





reservoir storage, land use and land cover changes) in altering the land-surface fluxes at various
spatio-temporal scales.

***Extension of Budyko's "supply and demand" Framework for Reservoir Operation and***
***Hedging***

We extend the Budyko framework for reservoir operation to meet the target demand

based on the standard operating policy (SOP) and linear hedging policy (Draper and Lund,
2004).  A hedging policy in reservoir operation aims to conserve water for future use by
curtaining the current demand (Draper and Lund, 2004).  Given an initial storage ($S_{t-1}$), inflow
($I_t$), demand ($D_t$) and evaporation ($E_t$) over a given time step ($t$), one could obtain the actual
release ($R_t$), and ending storage ($S_t$) along with spill ($SP_t$) using a simple mass balance (equation

1).

$$S_t = S_{t-1} + I_t - E_t - R_t - SP_t \qquad \qquad \dots (1)$$

By defining available water, $AW_t = S_{t-1} + I_t - E_t$, we obtain release (as "actual") under a given
hedging fraction ($0 \le \alpha \le 1$) for three reservoir storage conditions using equation 2. The SOP of a
reservoir simply corresponds to $\alpha = 1$ by supplying available water or demand at a given time.
$$\begin{aligned} S_t &= S_{\max}, R_t = D_t, SP_t = AW_t - D_t - S_{\max} && \text{if } AW_t - D \ge S_{\max} \\ S_t &= AW_t - R_t, R_t = \alpha D_t, SP_t = 0 && \text{if } S_{\min} < AW_t - D_t < S_{\max} \qquad \dots (2) \\ S_t &= S_{\min}, R_t = AW_t, SP_t = 0 && \text{if } S_{\min} \le AW_t - D_t \end{aligned}$$

Rewriting $AW_t$ as "supply", $D_t$ as "demand" and $R_t$ ("actual"), we develop the Budyko
framework for the reservoir operation under SOP and hedging policy (Figure 4). The SOP
simply provides the asymptotes, the upper bounds, for the $R_t / AW_t$ ("actual"/ "supply") ratio.
Figure 4 also demonstrates the developed framework for a hypothetical system for estimating the
monthly releases (see supporting information (SI) Tables 1-2 for data and details). Increased
hedging reduces the release and increases the storage and spill from the system. For
demonstration, a linear hedging policy is applied. But the real-world system operation will have
a complex non-linear release policy, still the data points are expected to lie within the bounds.
For systems with a small storage-to-demand ratio, the spill portion on the left asymptote is
expected to be much longer than a system with large storage-to-demand ratio. Similarly, for
systems with large (small) storage-to-demand ratio, most data points are expected to lie below
(on) the asymptotes portion of the framework. Given that this framework in Figure 4 is non-
dimensional, we could analyze release to demand characteristics for reservoirs with competing
purposes (e.g., hydroelectric vs flood control) and synthesize how release patterns vary based on
the demand-to-available water ratio across different type of systems.  Similarly, one can also
formulate the functional forms for non-linear hedging policy like Budyko equations as the upper
bounds are specified by the "supply and demand" relationship.

***Representing Human Demand and Environmental Flows from Reservoir Operation***

Reservoir storages reduce the runoff variability to meet the human demand, thereby

resulting in significant flow alterations (Wang et al., 2014). By adding a dedicated term,
environmental flow, $EF_t$, we rewrite the reservoir mass balance in equation (3).
$$S_t = AW_t - R_t - EF_t - SP_t \qquad\qquad \dots (3)$$
Given our variable of interest here is $EF_t$ ("actual"), we represent the "demand" as $R_t + EF_t$ and
available water, $AW_t$, as "supply", which gives us a simple framework to visualize the ratio,
environmental flow allocation $EF_t / AW_t$, has the upper bound $AW_t$, which is specified by the



1:1 line.  The term, $1 - EF_t / AW_t$, simply represents the alteration ratio at a given time step. The
lower bound specifies only allocation ( $R_t / EF_t = 0$ ) for human demand and a slope of 0.5
indicates equal allocation for human need and ecological demand. For instance, if $R_t / EF_t$ falls
below the slope of 0.5, it indicates significant flow alteration to meet human demand. In the case
of Falls Lake (Figure 5), a major water supply reservoir in the triangle area in NC (see SI Table 3
for data and additional details), it is evident that flow alteration is significant due to increased
allocation for human demand since more data points lie below the equal allocation line.  Using
the proposed framework in Figure 5, one could synthesize how reservoir systems with large
residence times, which is otherwise known as degree of regulation, impact flow alteration under
arid and humid conditions. The negative linear trend indicates (Figure 5) increased allocation
human use results in decreased environmental flow allocation. For instance, reservoirs in arid
(humid) climates are typically larger to reduce the larger (smaller) interannual variability in
runoff, hence such systems are expected to have higher (lower) degree of regulation. However,
this synthesis of reservoir systems across different climatic regimes needs to be evaluated in the
context of withdrawal for human use and their purpose and the consumptive use associated with
it. We argue the proposed framework could be useful for understanding the trade-off between
water allocation for human use and downstream ecological requirements.

***Interaction between Evapotranspiration and Sensible Heat***

Land use and land cover changes due to urbanization modify the evapotranspiration due

to limited water availability resulting in increased differences between urban and rural
temperature during the nighttime, which creates an urban heat island. Expressing the net
radiation, $R_n$, as the "supply" of energy available at the surface,  the latent heat flux (*LE*) as the





"demand",  and the sensible heat flux, *H*, as the "actual" variable of interest, we developed the
bounds (Figure 6) between the latent heat flux ratio ($LE/R_n$) and the sensible heat flux ratio
($H/R_n$). The basis for considering the latent heat flux as the "demand" stems from the view that
net radiation is effectively utilized for evapotranspiration in regions with increased water
availability with the residual energy being converted to net sensible heat flux.  For the hourly
data presented in Figure 6, latent heat flux indirectly quantifies the available soil water. The
proposed framework in Figure 6 could also be obtained by representing the evapotranspiration
ratio (Figure 1) as latent heat ratio with latent heat as "actual", net radiation as "supply" and
potential evapotranspiration as latent heat capacity (i.e., "demand"). Given Figure 6, one could
use this framework to evaluate the differences in sensible heat flux between urban and rural
settings by comparing across regions with abundant and limited water availability.  Figure 6
evaluates the proposed framework by plotting the hourly (7 AM- 5 PM) climatology of latent
heat flux ratio and sensible heat flux ratio in August from two FLUXNET towers
((https://fluxnet.fluxdata.org/), one from the urban setting and another from the rural setting, near
Minneapolis, MN.  The hourly climatology of H, LE and Rn, show the urban tower experience
more sensible heat than the rural tower during the daytime (Figure SI-1). However, the primary
challenge in using the FLUXNET data for evaluating the framework is due to the non-
availability of FLUXNET towers in urban settings. Identifying pairs of FLUXNET stations in
urban and rural settings and synthesizing the differences in urban and rural temperature under
different climatic regimes would provide us a pathway to understand the urban heat island effect.
Information available on the infrastructure characteristics and the type of pavement could also be
useful in explaining the spatial variability in the difference between urban and rural temperature.
Understanding how the sensible heat flux varies between urban and rural regimes across





different hydroclimatic regimes (i.e., arid vs humid) as the water availability in the urban
landscapes control the sensible heat.

We argue Budyko's supply and demand framework should not be considered just for

long-term water balance. As the supply and demand framework is based on conservation
equations, it could be exploited for understanding and quantifying the spatial variability in land-
surface fluxes under natural and human-altered landscapes. Figures 4-6 provide an extension of
the Budyko framework for understanding how land-surface fluxes are modified due to human
influence. Understanding the key drivers that alter the spatial variability of land-surface fluxes
using the modified and extended Budyko's framework should help in identifying the relevant
low-dimensional attributes that control the regional hydroclimate of human-altered
watersheds/landscapes. For long-term ET, it is the aridity index. For infiltration, it is the ratio of
infiltration capacity to rainfall depth. For reservoir operation, it is the ratio of human water
demand to available water in reservoir. For environmental flows, it is the competition with
human demand and available water. For the urban heat island, it is the water availability that
suppresses the sensible heat due to evaporative cooling. Thus, the low-dimensional attribute
varies for each environmental issue. Further, extending Budyko's framework for such
anthropogenic causes should enable the explicit decomposition and attribution of changes in
land-surface fluxes at various temporal scales resulting from changes in local/regional
hydroclimate or watershed-level modification. To refine existing hydroclimatologic models and
datasets developed at the regional, continental, global scale, a synthesis study is neeeded to
understand how the land-surface response varies across natural and human-altered watersheds.
Such a synthesis effort is also expected to enable a systematic decomposition of watershed-scale



anthropogenic influences and large-scale climate impacts in modulating land-surface fluxes at a
global scale, providing a tribute to Budyko's legacy.

**Opportunities, challenges, and relevance to other hydrologic synthesis studies**

Emphasis on understanding the complex interactions and feedback between human and

hydrological systems has renewed focus on "Socio-hydrology" (Sivapalan et al., 2012). The
impact of water use, land use and land cover and other anthropogenic influences on watershed
runoff and the associated non-stationary issues have been referred to as the study of "Hydro-
morphology" (Vogel, 2011). Vogel et al. (2015) argue that "to resolve the complex water
problems that the world faces today, nearly every theoretical hydrologic model introduced
previously is in need of revision to accommodate how climate, land, vegetation, and
socioeconomic factors interact, change, and evolve over time." Study of the interaction between
humans and the earth system has also received considerable support from various agencies such
as the National Science Foundation, the National Institute of Food and Agriculture and the U.S.
Geological Survey with targeted programs (e.g., Water Sustainability and Climate, Coupled
Human-Natural Systems and Innovations in Food-Energy-Water Systems, NAQWA). Thus,
evolving the Budyko framework to understand how land-surface responses vary under natural
and human-altered landscapes will also support various ongoing studies on the impact of human
influence on hydrological systems.

Enhancements to the Budyko framework will also support other ongoing activities that

focus on improving the ability to predict the hydrologic behavior of natural and ungauged
watersheds. As competition for water has increased, there has been increasing attention placed
on the need for water availability information at ungauged locations, even in regions where water
has not been considered in the past to be a limited resource. For these reasons, the decade from





2003 to 2012 was recognized by the International Association of Hydrological Sciences as the
Prediction in Ungauged Basins (PUB) Decade (Sivapalan et al., 2003). Blöschl et al. (2013;
tables A7-A10) showed that several methods to predict streamflow in ungauged watersheds have
been proposed; however, no one method has been universally accepted or demonstrated to work
in all hydrologic settings. Other studies have evaluated predictability in ungauged basins at the
global scale (Hrachowitz et al., 2013). Since the Budyko framework provides an approach for
improving our understanding of ungauged basins, there is potential cross-fertilization in various
ongoing studies for evaluating the extended Budyko framework and datasets for supporting
various global- and continental-scale hydrologic initiatives.

Another exciting aspect of the extension of the Budyko framework for considering

anthropogenic influences, involves the development of hydrologic indicators for a wide range of
purposes ranging from watershed classification, environmental permitting and a variety of water
management activities.  There is a continuing need to develop hydrologic indicators which are
founded in the science of hydrology, for the purpose of watershed classification as expressed so
nicely by Wagener et al. (2007).  The idea of plotting nondimensional variables, analogous to the
nondimensional variables proposed in Figure 1, has a very close association with the
development of nondimensional hydoclimatologic indicators for both natural (Weiskel et al.
2014) and human dominated (Weiskel et al., 2007) watershed systems.  For example, the aridity
and runoff ratios, two commonly used nondimensional hydroclimatic indicators arise naturally
from the Budyko framework for natural watersheds.  We anticipate that a wide range of new
hydrologic indicators, founded on the science of hydrology, yet useful for water management
and watershed classification, will arise from the types of studies envisioned here which extend
the Budyko framework to accommodate anthropogenic influences.



One significant challenge in evaluating the Budyko framework under human-altered
landscapes would be the availability of data on hydroclimate, storages, and human influences -
water withdrawal and land use changes, reservoir storages and releases - at different spatio-
temporal scales.  The monthly change in total water storage is a critical component of accurate
assessments of land-surface fluxes particularly in regions of high anthropogenic influence where
storage is impacted by pumping of groundwater resources, or conversion of surface water to
evapotranspiration through diversion for irrigation. In addition to the tremendous challenges
relating to data availability, there is the open research question of how we can capture the
complexity of human-water systems with a low dimensional parsimonious modeling approach.
One approach involves a gradual refinement of model features – a top-down approach – as
needed (Zhang et al., 2008; Sivapalan et al., 2003). Another strategy involves development of
critical data sets and then addition of model features as the spatio-temporal scale of the data
permits.  Such a global synthesis effort will require sources of several global-scale data sets from
a variety of sources, including remotely sensed data. The selection of appropriate data at this
scale presents challenges in balancing spatial resolution and uncertain accuracy and consistency
among the considered data sets.  Findings from another synthesis study titled, "Water
Availability for Ungaged Basins" revealed that, as various hydrologic modeling communities
converge towards continental-domain hydrologic models, these communities will encounter
similar limitations and challenges (Archfield et al., 2015).  It is our hope and contention that the
Budyko framework can provide a unifying perspective for bridging gaps in hydrologic data
availability and model resolution over a wide range of spatial and temporal scales. As shown in
this opinion article, the framework can also be modified beyond the traditional long-term balance





for understanding how the land-surface responses, runoff and evapotranspiration, vary across
natural and human-altered landscapes.

**Acknowledgements:** This research was funded in part under award EAR- 1823111from the
National Science Foundation (NSF) and from the United States Geological Survey (USGS)
Powell Center Working Group Project "A global synthesis of land-surface fluxes under natural
and human-altered watersheds using the Budyko framework".  The data used in this paper is
provided in the supplemental information.

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








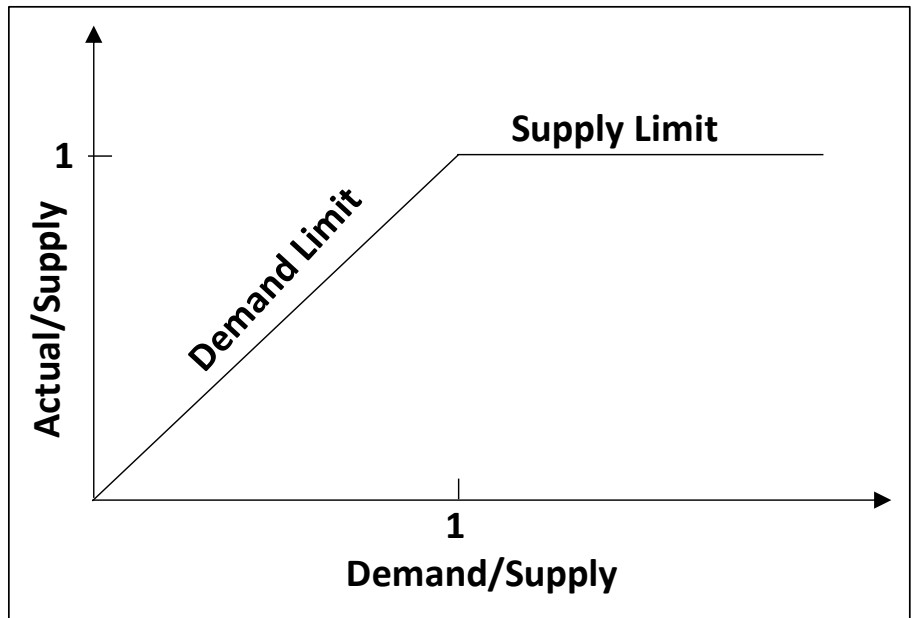

Figure 1: An overview of the Budyko supply and demand framework for understanding the land-
surface flux response (actual) over natural and human-altered watersheds. The "limits" concept
as suggested by Budyko (1958) quantifies the actual response (Y axis) based on the physical
demand-to-supply ratio of energy/moisture over the control volume or the watershed.



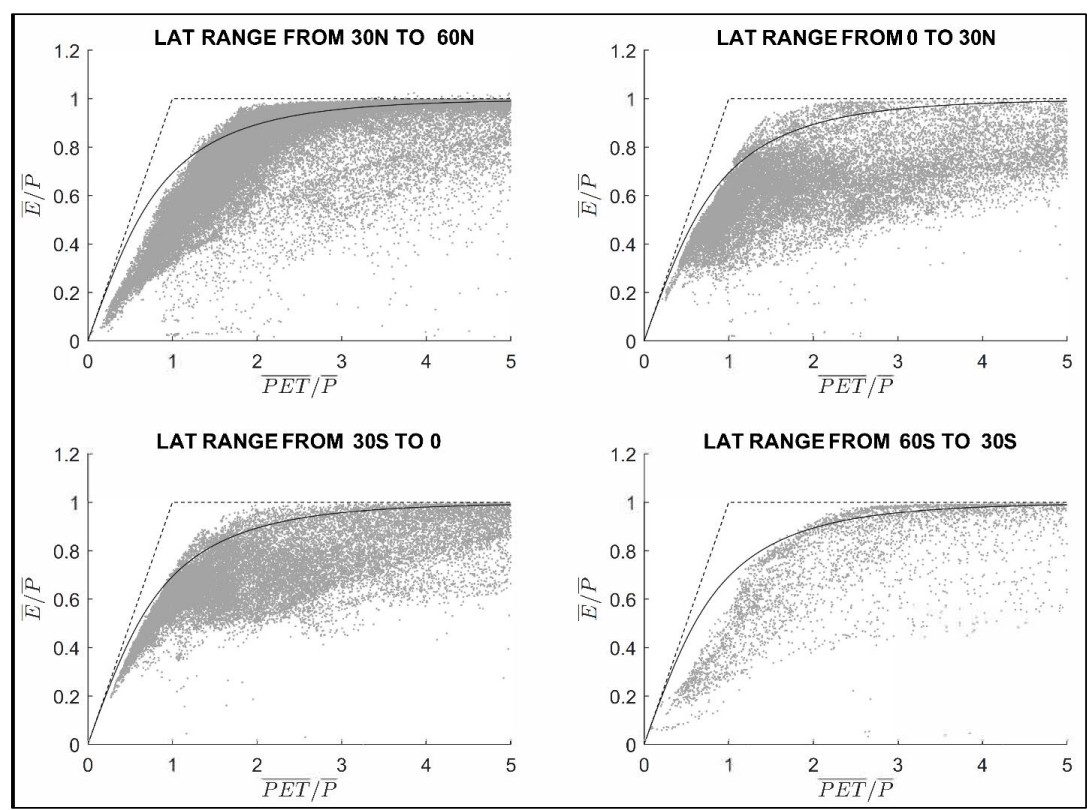



Figure 2: The traditional Budyko framework for long-term water balance along with the asymptotes and the Budyko curve ($\overline{ET}/\overline{P}=\left[\left(1-\exp(-\overline{PET}/\overline{P})\right)*\overline{PET}/\overline{P}*\tanh(\overline{PET}/\overline{P})^{-1}\right]^{0.5}$). The ratio of mean annual potential evapotranspiration ($\overline{PET}$, demand) to mean annual precipitation ($\overline{P}$, supply) explains the ratio of mean annual evapotranspiration ($\overline{ET}$, actual) and $\overline{P}$, and the data points are from GLDAS-2 estimates at the pixel level (0.25 º) for the period 1948-2010 over the northern (top row, 0º-30º and 30º-60º latitudes) and southern (bottom row, -30º to 0º and -30º to -60º latitudes) hemispheres.




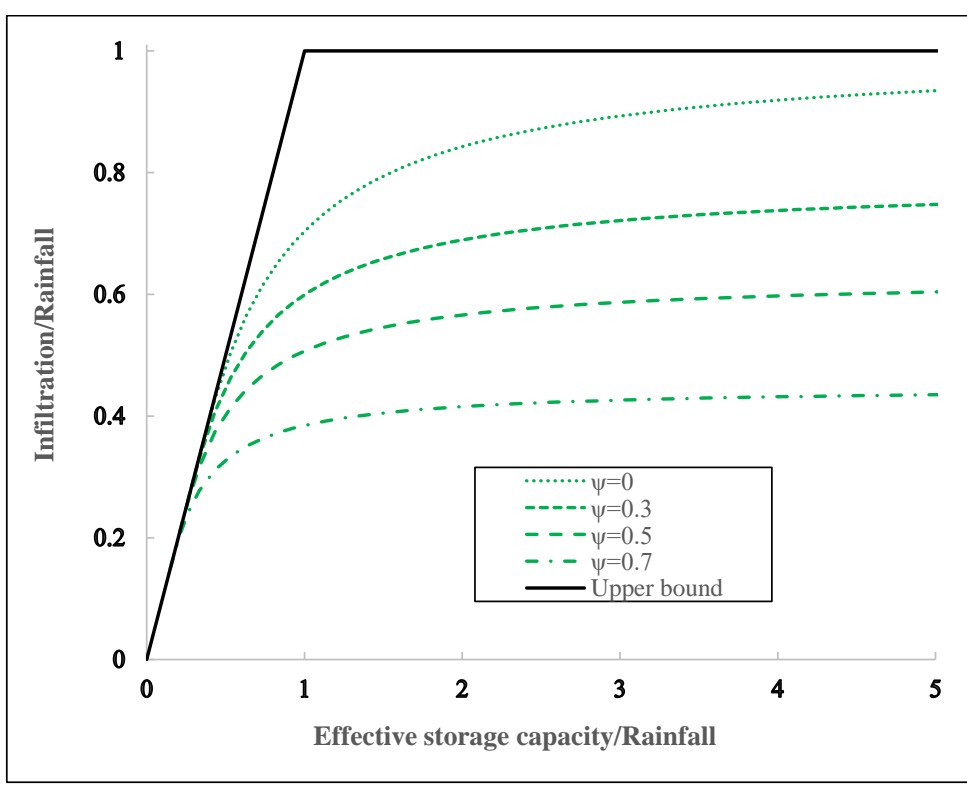


Figure 3: Modeling infiltration in the Budyko's supply and demand framework: the ratio of infiltration (actual) and rainfall depth is a function of the ratio of infiltration capacity (demand)and rainfall depth (supply) as well as the initial soil moisture condition represented by the degree of saturation ($\psi$) (Reproduced from Wang (2018)).







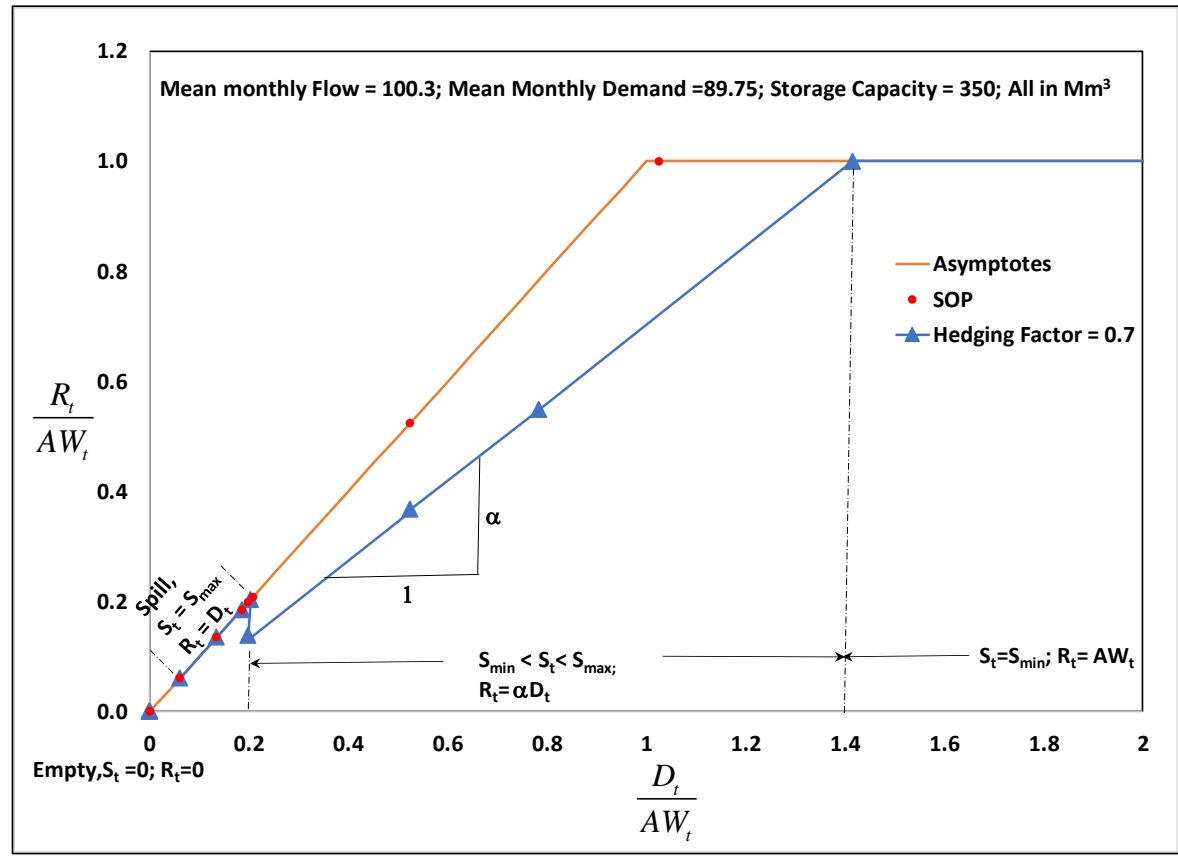


Figure 4: Modeling hedging policy of reservoir operations in the Budyko's supply and demand
framework. The standard operating policy (SOP) is corresponding to the asymptotes. For the
hedging rule, delivery or release is "actual", available water is "supply", and human use is
"demand". For demonstration purpose, a linear function is assumed for the hedging rule (i.e.,
$R_t = \alpha D_t$). The storage conditions are indicated for the hedging policy alone.

570

571

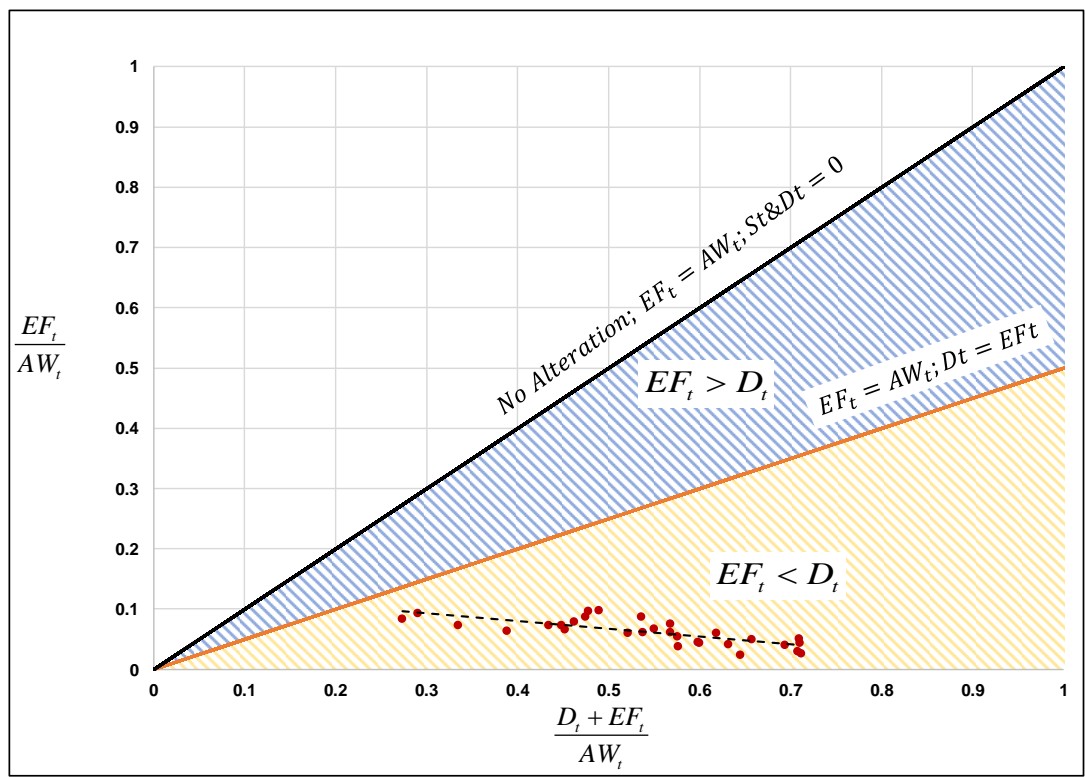

572

Figure 5: Modeling synthesizing flow alteration in the Budyko's supply and demand framework:
the ratio of environmental flow ("actual") and the available water is a function of the ratio the
total demand for human and environmental flow ("demand") and the available water ("supply").
Annual flows from Falls Lake (red dots) show human withdrawal for water supply is more than
the downstream environmental flow release.

578

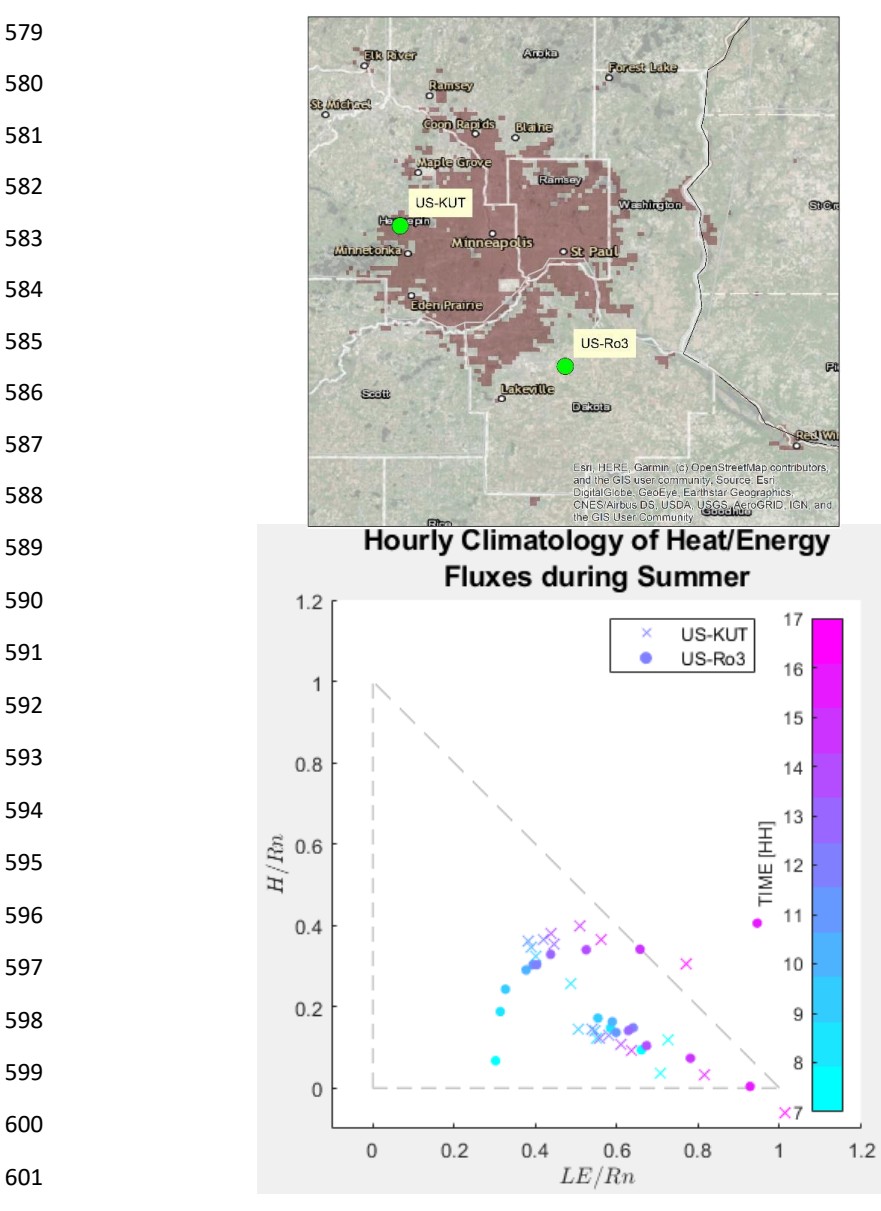

Figure 6: Extending the Budyko framework (bottom figure) for quantifying the sensible heat ("actual") based on available energy ("supply") and latent heat ("demand") for two FLUXNET towers (top figure), US-KUT and US-RO3, from an urban area (brown shaded) and rural area (green shaded). The ratio of mean hourly sensible heat to mean hourly net radiation is plotted against the ratio of mean hourly latent heat to the mean hourly net radiation from the two towers. Figure SI-1 compares the average hourly values from 7 AM to 5 PM for August 2006 and 2007.
© OpenStreetMap contributors 2019. Distributed under a Creative Commons BY-SA License.