# Peer review of "HESS Opinions: Beyond the Long-term Water Balance: Evolving Budyko's Legacy for the Anthropocene towards a Global Synthesis of Land-surface Fluxes under Natural and Human-altered Watersheds"

_Hydrology and Earth System Sciences, 2019_

## Short Comment (SC1) · 19 Sep 2019

Here are some thoughts/queries that I had while reading the paper (that by sharing hopefully strengthen the revised version of this paper). This is not a complete review (since I am not assigned being a reviewer). I enjoyed reading the paper.

- It is stated that Budyko has been verified "over thousands of natural watersheds around the globe". However, the studies that you cite are not necessarily at the wa-

tershed scale (e.g. Williams is a FLUXNET based study, using plot scales), nor do I expect that all the thousands of watersheds in the other studies can be classified as "natural". In addition, what does "verified" really mean here? (Note hereby that e.g. many of Williams points fall outside the energy and water limits; Sivapalan does not present any Budyko curve in its study (only related concepts)).

- It is stated that there "...is a critical need to enable a complete understanding of global hydroclimate during the Anthropocene. The Budyko framework provides an ideal approach for such inquiry..." Prior to this statement, many aspects of change are listed, including flood changes. Listing this example, and stating you want a "complete" understanding of hydroclimatic change suggests to me that it should include changes in floods as well. What is your logical basis for using Budyko for understanding flood changes since its original use and assumptions have very little to do with hydrology at the short time-scales over which many floods are produced?

- It is stated that "Studies have also focused on the impact of land cover and climate on long-term water yield using global data (Zhou et al., 2015)". However, this study is mathematically flawed; see https://www.nature.com/articles/ncomms14795. This makes me question if this study is a good example to cite...

- In the section "Long term water balance" (line 106 and onwards) the ability of the original Budyko curve is tested in explaining global water balance variability. However, the ET data to which it is compared is model output. Would it make sense to use something more directly observation-based, to avoid that it remains unclear to what extent scatter around the curve is based on real-world behaviors, and to what extent it arises from inaccuracies of GLDAS? The fact that (almost?) not a single data point with aridity $<\sim 1$ plot above the Budyko curve is hereby interesting since this is not typically observed in other datasets (as far as I am aware).

- It is stated that "[...] limited/no effort has been undertaken on how this data cloud of long-term water balance cloud is expected to change under potential climate change

and how this interplay between moisture and energy is expected to affect the long-term water balance under different type of watersheds (Creed et al., 2014)." However, at the same time, your paper states that Budyko can be straightforwardly used to decompose the effects of climate change vs human influences (e.g. line 90-94), which also implies that you can straightforward use it to predict... This seems to be somewhat contradicting?

- The discussed "Extension of Budyko's "supply and demand" concept for infiltration" (and other suggested extensions) sounds interesting. However, we need to be aware that plotting variables using demand & supply axes that BOTH have the same term in their denominator partly show strong correlations/patterns because they have spurious self-correlations due to a common denominator [Bensen, 1965; Brett, 2004]. This does not mean we should not use it, I just think the community sometimes forgets about this fact (for example, it's rarely acknowledged that Budyko itself is partly a spurious self-correlations due to a common denominator).

References Benson, M. A. (1965). Spurious correlation in hydraulics and hydrology. Journal of the Hydraulics Division, 91(4), 35-42. Brett, M. T. (2004). When is a correlation between non‐independent variables "spurious"?. Oikos, 105(3), 647-656. Gudmundsson, L., Greve, P., & Seneviratne, S. I. (2017). Correspondence: Flawed assumptions compromise water yield assessment. Nature communications, 8, 14795.

---

## Referee Comment (RC1) · Anonymous Referee #1 · 24 Oct 2019

**Review of HESS Manuscript #2019-418**

Title:     HESS Opinions: Beyond the Long-term Water Balance: Evolving Budyko's Legacy
          for the Anthropocene towards a Global Synthesis of Land-surface Fluxes under
          Natural and Human-altered Watersheds

Author:   Sankarasubramanian et al.

**Review**

This proposed HESS opinion manuscript describes how the Budyko-supply-demand type analysis can be extended to cover several other important problems in Hydrology.

The manuscript is well written and plays a useful opinion/review type role in the literature.

I had few comments.

The main substantive scientific comment relates to Figure 6 and the associated text. In short, I did not understand the point. The basic ideas for partitioning Rn between LE and H over different surface types have been covered in 1000s of papers. The urban example used (in Fig. 6) is also covered extensively in the classic text on urban climatology by Prof. Oke. In short, Fig. 6 and the associated text really does not belong in this manuscript.

**Recommend: Accept after (minor) revisions.**

**Comments**

**1.**     Lines 118-120. Please explain how your PET has been calculated.
**2.**     Line 168. "simplistic" is the wrong word I think. simple might be a better choice.
**3.**     Line 191. TYPO. curtailing
**4.**     Line 193. delete simple (not needed).

---

## Referee Comment (RC2) · Anonymous Referee #2 · 27 Oct 2019

**Review of** *"HESS Opinions: Beyond the Long-term Water Balance: Evolving Budyko's*
*2 Legacy for the Anthropocene towards a Global Synthesis of Land-surface*
*3 Fluxes under Natural and Human-altered Watersheds"* **by  Sankarasubramanian et al.**

The article written by Sankarasubramanian et al., considers the Budyko-framework and the authors explain the framework in a demand-supply setting. They give several examples relating to infiltration, reservoir operations, environmental flows and sensible and latent heat fluxes. The authors argue that formulating especially anthropogenic influences in a demand-supply framework will help explain land-surface responses.

I think the article is very interesting. I fully understand the point of the authors, and I like the idea to use simple demand-supply approaches to understand complex system responses. The article is also well written, and I only have several relatively minor comments.

The proposed method strongly depends on what is defined as demand, and what is defined as supply. For example, in the Budyko-framework one could define the supply in terms of precipitation (i.e. normalize by the precipitation), but also by the potential evaporation. Placing the data in different projections of supply and demand will probably lead to different interpretations, and I wondered if the authors have any suggestions on how to do this systematically.

In addition to this, the authors argue that putting variables in a demand-supply framework will help to understand also spatial-temporal variability. However, the definitions of supply and demand will change depending on the time scale or the spatial scale. For example, the Budyko-framework can be applied on longterm-data for large catchments as the storage term in the water balance and small scale spatial variations (i.e. extractions, leaky catchments) become negligible, but as timescales change, the definitions of  demand and supply will probably change too.

I wonder if the example in Figure 2 is appropriate.  The Budyko-framework generally works best for larger basins, whereas here a 0.25 degree resolution is used. More small scale variation might come in, which is often not particularly well-handled in land surface models. For lower values of Ep/P the data also seems to plot always rather far below the analytical curve. I know the authors just want to show an example here, but the spread of the data could also just be explained by model deficiencies in this case. It may better to use observed data, for example the Camels-dataset might be of use for the authors.

The last example seems a bit off in my view. Generally, "actual" is a realization (or a percentage) of the "demand", varying between 0 and 100%. In this example this is not the case, sensible heat is not a percentage of the latent heat. I think, and the authors mention it themselves on page 12, lines 254-257, putting the framework in terms  LE as "actual", Rn as "supply" and potential evaporation as "demand", makes more sense and is more in line with the other examples.

Related to this, and actually also my first points, I think it might help to elaborate in the general description of the approach (Budyko Framework Adaptation in  Watershed Modelling, p5, line 95) on how to define supply and demand in a consistent way. I my view, the "actual" realization should always be a certain percentage of the "demand", but I'd like to know what the authors propose here.

My last point is merely just a suggestion from my side, but the title does not seem to fully capture the content.  At first, I expected (another) extension of the analytical equations in the Budyko framework with a new term accounting for the anthropogenic impact, similar to how some of the cited studies

included soil moisture or seasonality in the analytical equations.  However, the authors make their argument a lot more general, which I really like, by simply using a demand-supply formulation for complex problems, and from which the Budyko-framework is actually just an example. I think it is important to add at least the supply-demand terms in the title.

Concluding, it is an interesting article that made me really think about how to use supply-demand formulations in hydrology. I hope my comments are useful for the authors and look forward to a new version of the manuscript.

**Minor comments**
P3.L47. Darwinian approach → The Darwinian approach?
P6.L129-130. Suggestion: see the review paper of Wang et al. (2016)
P6.L131-132. Suggestion: this cloud of long term water balance data
P6.L133. Please note, it's not only energy and soil moisture, the vegetation and thus the transpiration will be affected too.
P9.Eq2. Is the third condition correct? Shouldn't it be when Smin > Aw – D   ?
P10.L207. Not sure if this is true, especially when hedging is applied, when there is storage, more water can theoretically be released than the demand. I would also expect that points would fluctuate around the line, due to delays in a target-feedback-loop, i.e. the actual hedging factor will never be reached exactly and always be slightly bigger or smaller compared to the operational rule. This is different compared to Budyko, where the limits are really physical limits, and not operation based limits.
P11.L235-236. Increased allocation human use → increased allocation for human use
P13.L275-276. For long-term water balance –> for the long-term water balance
Figure 5. I suggest to use a colorscale for the yearly points, so see if there is a temporal trend.
Figure 6. This figure looks like a matlab-screenshot, I would suggest to format the plot a bit nicer.

---

## Author Comment (AC1) · 9 Jan 2020

Response to open review comments by Dr. Wouter Berghuijs

Thanks for taking the time to review our paper. Here is our detailed response to your comments.

- It is stated that Budyko has been verified "over thousands of natural watersheds around the globe". However, the studies that you cite are not necessarily at the watershed scale (e.g. Williams is a FLUXNET based study, using plot scales), nor do I expect that all the thousands of watersheds in the other studies can be classified as "natural". In addition, what does "verified" really mean here? (Note hereby that e.g. many of Williams points fall outside the energy and water limits; Sivapalan does not present any Budyko curve in its study (only related concepts)).

Response: We have modified the sentence. Changed "thousands" to "numerous". Removed Williams et al., 2012; Sivapalan et al., 2011. Added Sankarasubramanian and Vogel (2003), Li et al., 2013.

- It is stated that there "... is a critical need to enable a complete understanding of global hydroclimate during the Anthropocene. The Budyko framework provides an ideal approach for such inquiry..." Prior to this statement, many aspects of change are listed, including flood changes. Listing this example, and stating you want a "complete" understanding of hydroclimatic change suggests to me that it should include changes in floods as well. What is your logical basis for using Budyko for understanding flood changes since its original use and assumptions have very little to do with hydrology at the short time-scales over which many floods are produced?

Response: In this context, we mention several hydroclimatological processes that have changed over time due to anthropogenic influences. Flood is one of them. Our argument from this paper is that Budyko's framework can be extended to even shorter time scales. For instance, Figure 3 presents the framework for infiltration, which is an event scale hydrological process. Flooding could be considered as the land-surface response in excess of infiltration. Thus, the idea is to propose possible novel extensions of the Budyko Framework so that low dimensional nature of the process and the associated drivers could be identified.

- It is stated that "Studies have also focused on the impact of land cover and climate on long-term water yield using global data (Zhou et al., 2015)". However, this study is mathematically flawed; see https://www.nature.com/articles/ncomms14795. This makes me question if this study is a good example to cite...

**Response:** We agree with this suggestion. We have removed this reference and added Li et al., (2013) and Wang and Tang (2014).

- In the section "Long term water balance" (line 106 and onwards) the ability of the original Budyko curve is tested in explaining global water balance variability. However, the ET data to which it is compared is model output. Would it make sense to use something more directly observation-based, to avoid that it remains unclear to what extent scatter around the curve is based on real-world behaviors, and to what extent it arises from inaccuracies of GLDAS? The fact that (almost?) not a single data point with aridity

---

## Author Comment (AC2) · 9 Jan 2020

Dear Editor

Thanks for handling our opinion paper. We are submitting the response for the reviewers' comments.

Sankar Arumugam

**Review of HESS Manuscript #2019-418**

Title:    HESS Opinions: Beyond the Long-term Water Balance: Evolving Budyko's Legacy for the Anthropocene towards a Global Synthesis of Land-surface Fluxes under Natural and Human-altered Watersheds

Author:   Sankarasubramanian et al.

**Review**

This proposed HESS opinion manuscript describes how the Budyko-supply-demand type analysis can be extended to cover several other important problems in Hydrology.

The manuscript is well written and plays a useful opinion/review type role in the literature.

I had few comments.

The main substantive scientific comment relates to Figure 6 and the associated text. In short, I did not understand the point. The basic ideas for partitioning Rn between LE and H over different surface types have been covered in 1000s of papers. The urban example used (in Fig. 6) is also covered extensively in the classic text on urban climatology by Prof. Oke. In short, Fig. 6 and the associated text really does not belong in this manuscript.

**Response to Reviewer-1's comments**

Response: Thanks for the review of our article. We agree with your comment on the established literature on the partitioning of latent heat and sensible heat. Our primary contribution here on Figure 6 is in presenting the partitioning of surface energy balance in the Budyko's supply and demand framework and also in demonstrating the observed fluxes from urban and nearby rural settings fall within the bounds as suggested by the Budyko's framework. Hence, we would like to keep the figure in the overall scope for extension of the framework. However, we have revised Figure 6. We have also clearly defined the "actual" (in Figure 6, it is sensible heat), supply ("net radiation") and demand ("latent heat").  So, figure 6 presents the budyko framework for urban climatology to understand urban heat island issues.

**Comments**

1.    Lines 118-120. Please explain how your PET has been calculated.
2.    Line 168. "simplistic" is the wrong word I think. simple might be a better choice.
3.    Line 191. TYPO. curtailing
4.    Line 193. delete simple (not needed).

Line 118-120:  ET and PET estimates were obtained from the NOAH Land Surface Model, which uses Penman-Monteith method for calculating the land-surface fluxes (Rui, 2011).

Lines 168, 181 and 193: All corrected.

---

## Author Response (AR1)

**Dear Editor**

Dear Dr. Sankarasubramanian,

I had a close look at your manuscript, the two reviews and your corresponding responses. In line with both reviewers and Wouter Berghuijs I think that the proposed opinion is valuable and deserves to be published in HESS. However, prior to this I strongly recommend addressing the main points of the reviews and of the scientific comment with great care within a round of revisions.

As per your suggestions and three reviews, we have submitted the revised manuscript. Thanks for handling our paper.

In line with reviewer 1 I recommend removal of figure 6, simply because H and LE are subject to different demands and both fluxes are driven by different physics! The purpose of LE is to deplete the saturation deficit of the atmospheric boundary layer (ABL) and establish a minimum free energy state. So the demand is the saturation deficit integrated over the ABL. The general purpose of H is the generally the same, establishing a maximum entropy state, but with respect to corresponding temperature gradient in the ABL. To me it does hence not make sense to link these two variables in the proposed extended Budyko framework.

We have removed Figure 6 in the interest of coming to a closure on this manuscript. I still feel that sensible heat and latent heat are inter-related and can be represented in the Budyko framework. We will pursue this in our future publications.

In line with Wouter Berghuijs I like the transfer of the Budyko concept to infiltration and storage. The soil demand for infiltration is however not the storage capacity but the storage deficit from thermodynamic equilibrium/ soil hydraulic equilibrium storage as recently argued in one of my own publications in HESS (Zehe et al. 2019). In this context I thank that you should consider the danger of introducing spurious correlations at little more seriously.

We have included your publication in the references related to infiltration example. We have also mentioned the issues related to spurious correlation. As suggested caveats by Dr. Berghuijs, we have mentioned arising from spurious correlation in the paragraph after Figure 5.

**References:**

Zehe, E., Loritz, R., Jackisch, C., Westhoff, M., Kleidon, A., Blume, T., Hassler, S. K., and Savenije, H. H.: Energy states of soil water - a thermodynamic perspective on soil water dynamics and storage-controlled streamflow, Hydrology And Earth System Sciences, 23, 971-987, 10.5194/hess-23-971-2019, 2019.

Thanks for handling the manuscript.

Sankar Arumugam

**Review of HESS Manuscript #2019-418**

Title: HESS Opinions: Beyond the Long-term Water Balance: Evolving Budyko's Legacy for the Anthropocene towards a Global Synthesis of Land-surface Fluxes under Natural and Human-altered Watersheds

Author: Sankarasubramanian et al.

**Review**

This proposed HESS opinion manuscript describes how the Budyko-supply-demand type analysis can be extended to cover several other important problems in Hydrology.

The manuscript is well written and plays a useful opinion/review type role in the literature.

I had few comments.

The main substantive scientific comment relates to Figure 6 and the associated text. In short, I did not understand the point. The basic ideas for partitioning Rn between LE and H over different surface types have been covered in 1000s of papers. The urban example used (in Fig. 6) is also covered extensively in the classic text on urban climatology by Prof. Oke. In short, Fig. 6 and the associated text really does not belong in this manuscript.

**Response to Reviewer-1's comments**

Response: Thanks for the review of our article. We have removed Figure 6 in the interest of coming to a closure on this manuscript. I still feel that sensible heat and latent heat are interrelated and can be represented in the Budyko framework. We will pursue this in our future publications.

**Comments**

- 1. Lines 118-120. Please explain how your PET has been calculated.
- 2. Line 168. "simplistic" is the wrong word I think. simple might be a better choice.
- **3.** Line 191. TYPO. curtailing
- 4. Line 193. delete simple (not needed).

Line 118-120: ET and PET estimates were obtained from the NOAH Land Surface Model, which uses Penman-Monteith method for calculating the land-surface fluxes (Rui, 2011). This has been added in the revised manuscript.

Lines 168, 181 and 193: All corrected.

**Response to Reviewer-2**

**Review of "HESS Opinions: Beyond the Long-term Water Balance: Evolving Budyko's 2 Legacy for the Anthropocene towards a Global Synthesis of Land-surface 3 Fluxes under Natural and Human-altered Watersheds" by Sankarasubramanian et al.**

The article written by Sankarasubramanian et al., considers the Budyko-framework and the authors explain the framework in a demand-supply setting. They give several examples relating to infiltration, reservoir operations, environmental flows and sensible and latent heat fluxes. The authors argue that formulating especially anthropogenic influences in a demand-supply framework will help explain land-surface responses.

I think the article is very interesting. I fully understand the point of the authors, and I like the idea to use simple demand-supply approaches to understand complex system responses. The article is also well written, and I only have several relatively minor comments.

The proposed method strongly depends on what is defined as demand, and what is defined as supply. For example, in the Budyko-framework one could define the supply in terms of precipitation (i.e. normalize by the precipitation), but also by the potential evaporation. Placing the data in different projections of supply and demand will probably lead to different interpretations, and I wondered if the authors have any suggestions on how to do this systematically.

**Response:** Thanks for your comments. Our extension of Budyko framework for three different problems, infiltration, hedging and flow alteration (dropped the sensible heat flux extension), provide the context for identifying the "demand" term systematically. The term "demand" provides the upper bound of the "actual" variable if the "supply" variable is unlimited. For instance, in the case of infiltration as "actual", the "demand" is the maximum infiltration capacity, which implies if the "supply" (i.e., rainfall) is unlimited. We identify the demand for each proposed extension in this fashion. We have added the above highlighted definition of demand in the manuscript.

I wonder if the example in Figure 2 is appropriate. The Budyko-framework generally works best for larger basins, whereas here a 0.25 degree resolution is used. More small scale variation might come in, which is often not particularly well-handled in land surface models. For lower values of Ep/P the data also seems to plot always rather far below the analytical curve. I know the authors just want to show an example here, but the spread of the data could also just be explained by model deficiencies in this case. It may better to use observed data, for example the Camels-dataset might be of use for the authors.

**Response:** The reason we did not use observed data in this Figure is because numerous studies have demonstrated the applicability of long-term Budyko's curves for observed datasets (Sankarasubramanian and Vogel, 2003; Abatzoglu et al., 2017 and others), hence we are not presenting it. Further, presenting the long-term water balance from the GLDAS2 dataset shows the performance of the Budyko curve over various latitudes. To make the latitudinal distribution of fluxes as per the general circulation cells, we have revised Figure 2 with grouped at 10° intervals. The key point from the figure is that there is a lower bound on the evapotranspiration ratio for each aridity index range, which emphasizes the need for other controlling factors such as seasonality of forcings (precipitation and temperature) and soil water holding capacity in influencing the long-term water balance. Further, these low ET ratio happens in the horse

latitudes (20-40 N), whereas high ET ratio happens in places with rising circulation cells (0 to 10 N and 50-60N). In the case of the southern hemisphere, organization of circulation cells do not strictly follow latitudinal patterns as the land surface being proximity to the ocean, which makes ET ratio varying substantially from the circulation patterns.

---

## Referee Report (RR1)

**2nd Review of HESS Manuscript #2019-418**

Title:     HESS Opinions: Beyond the Long-term Water Balance: Evolving Budyko's Legacy for the Anthropocene towards a Global Synthesis of Land-surface Fluxes under Natural and Human-altered Watersheds

Author:   Sankarasubramanian et al.

**Review**

This is the 2nd review the above-named manuscript. I was reviewer 1 in the first reviews.

In preparing this review I have read the response to both editorial and review comments.

The authors have responded appropriately.

I noticed in the author_response I downloaded there was a missing responses at the end of the manuscript. These are not of scientific interest and can be cleaned up in the editing stage.

**Recommend: Publish.**

---

## Referee Report (RR2)

Review of **HESS Opinions: Beyond the Long-term Water Balance: Evolving Budyko's Supply-Demand Framework for the Anthropocene towards a Global Synthesis of Land-surface Fluxes under Natural and Human-altered Watersheds** by Sankarasubramanian et al.

The revised manuscript on using simple supply-demand frameworks in hydrology really showed some nice improvements compared to the previous version. I am happy the authors addressed my comments and I just have several minor issues.

Regarding Figure 2, I do understand what the authors want to show and I also agree with the authors that variations might be explained by other controlling factors. However, you cannot deny that the used model may have deficiencies as well, and I think this needs to be added at least in the discussion. So I suggest to just add some lines on that.

With regard to my comment on Figure 5, I meant adding a colorscale for each year (e.g. darker colors for more recent years). Mainly because I wondered if the points in Figure 5 are sequential. In other words, are the points towards lower ratios for more recent years and higher ratios for years more in the past? Is the environmental flow partition reducing over time?

From line 256, the authors start a more general discussion, which does not seem to fit under the current section of *Representing Human Demand and Environmental Flows in from Reservoir Operation.* I suggest to place this part in a different section.

I hope the authors find these rather minor issues helpful again. The manuscript is interesting and I recommend publication after these small issues have been addressed.

**Minor comments**
L236-237. Sentence seems a bit off.
L.541. Y axis → Y-axis
L.551-552. I think the descriptions of the top row and bottom row do not match with what the figure shows.

---

## Author Response (AR2)

Response to Reviewer-2's comments

The revised manuscript on using simple supply-demand frameworks in hydrology really showed some nice improvements compared to the previous version. I am happy the authors addressed my comments and I just have several minor issues.

Response: Thanks for your comments and time.

Regarding Figure 2, I do understand what the authors want to show, and I also agree with the authors that variations might be explained by other controlling factors. However, you cannot deny that the used model may have deficiencies as well, and I think this needs to be added at least in the discussion. So I suggest to just add some lines on that.

Response: The following text has been added: The evapotranspiration ratio plotted in Figure 2 could have bias as they are based on Noah land surface model estimates from GLDAS-2 model. For large basins, estimating evapotranspiration as the difference between precipitation and streamflow is more accurate as the ET ratio and aridity index are purely based on observed information (Sankarasubramanian and Vogel, 2003).

With regard to my comment on Figure 5, I meant adding a color scale for each year (e.g. darker colors for more recent years). Mainly because I wondered if the points in Figure 5 are sequential. In other words, are the points towards lower ratios for more recent years and higher ratios for years more in the past? Is the environmental flow partition reducing over time?

Response: There is no temporal pattern as the data points are more influenced by inflow variability. So, we did not change the colors of those data points. However, there were few minor edits on the labeling with some of the subscripts were not proper. Hence, we revised Figure 5.

From line 256, the authors start a more general discussion, which does not seem to fit under the current section of Representing Human Demand and Environmental Flows in from Reservoir Operation. I suggest to place this part in a different section.

Response: We revised it as follows: We argue that the Budyko supply and demand framework could also be considered for understanding the role of humans in altering the land-surface fluxes.

Minor comments

L236-237. Sentence seems a bit off.

Response:

L.541. Y axis → Y-axis

Response: Modified as suggested.

L.551-552. I think the descriptions of the top row and bottom row do not match with what the figure shows.

Response: Modified as suggested.